# Expanding the Clinical Spectrum of *CEP290* Variants: A Case Report on Non-Syndromic Retinal Dystrophy with a Mild Phenotype

**DOI:** 10.3390/genes15121584

**Published:** 2024-12-09

**Authors:** Anna Esteve-Garcia, Cristina Sau, Ariadna Padró-Miquel, Jaume Català-Mora, Cinthia Aguilera, Estefania Cobos

**Affiliations:** 1Clinical Genetics Unit, Metropolitan South Clinical Laboratory, Bellvitge University Hospital, Institut d’Investigació Biomèdica de Bellvitge (IDIBELL), L’Hospitalet de Llobregat, 08907 Barcelona, Spain; aesteveg@bellvitgehospital.cat (A.E.-G.); cristinasau@bellvitgehospital.cat (C.S.); 2PhD Program in Genetics, Faculty of Biology SED of Medicine and Health Sciences, University of Barcelona, 08028 Barcelona, Spain; 3Genetics Laboratory, Metropolitan South Clinical Laboratory, Bellvitge University Hospital, Institut d’Investigació Biomèdica de Bellvitge (IDIBELL), L’Hospitalet de Llobregat, 08907 Barcelona, Spain; apadro@bellvitgehospital.cat; 4Department of Ophthalmology, SJD Barcelona Children’s Hospital, Esplugues de Llobregat, 08950 Barcelona, Spain; jaume.catala@sjd.es; 5The Hereditary Retinal Dystrophies Unit, Ophthalmology Departments at SJD Barcelona Children’s Hospital and University Hospital of Bellvitge, 08950 Barcelona, Spain; 6Department of Ophthalmology, Bellvitge University Hospital, Institut d’Investigació Biomèdica de Bellvitge (IDIBELL), L’Hospitalet de Llobregat, 08907 Barcelona, Spain

**Keywords:** *CEP290*, retinal dystrophy, phenotypic heterogeneity

## Abstract

**Background/Objectives**: Biallelic pathogenic variants in the *CEP290* gene are typically associated with severe, early-onset inherited retinal dystrophies (IRDs) in both syndromic and non-syndromic forms. This study explores the phenotypic variability of non-syndromic IRDs associated with *CEP290* variants, focusing on two siblings with biallelic variants, one of whom exhibits a remarkably mild phenotype, thereby expanding the clinical spectrum. **Methods**: Whole-exome sequencing (WES) and mRNA analysis were performed to identify and characterize *CEP290* variants in the siblings. Comprehensive ophthalmologic evaluations assessed retinal function and disease progression. **Results**: Two *CEP290* variants, a frameshift (c.955del, p.(Ser319LeufsTer16)) and a missense (c.5777G>C, p.(Arg1926Pro)), were identified in *trans* in both siblings. Despite sharing the same genetic variants, the sister exhibited significantly preserved retinal function, while the brother presented with a more severe, progressive retinal dystrophy. **Conclusions**: This study broadens the phenotypic spectrum of non-syndromic *CEP290*-related IRDs, demonstrating variability in disease severity ranging from mild to severe. These findings highlight the importance of personalized monitoring and tailored management strategies based on individual clinical presentations of *CEP290*-related IRDs.

## 1. Introduction

The *CEP290* gene encodes a protein essential for ciliary function, particularly in photoreceptor cells, where it is critical for maintaining the structure and function of the connecting cilium [1,2]. This structure facilitates protein and vesicle transport, which are vital for photoreceptor activity and overall retinal function [3]. While *CEP290* is expressed in various tissues, including blood and fibroblasts, its expression is particularly enriched in photoreceptors, reflecting its pivotal role in this tissue [3,4]. Pathogenic variants in *CEP290* are associated with a wide spectrum of ciliopathies, including syndromic conditions, such as Joubert syndrome (JBTS, MIM#610188), Senior–Loken syndrome (SLSN, MIM#610189), and Meckel syndrome (MKS, MIM#611134), as well as non-syndromic inherited retinal dystrophies (IRDs), like Leber congenital amaurosis (LCA, MIM#611755), cone-rod dystrophy (CRD), and retinitis pigmentosa (RP) [5,6,7,8,9,10].

Historically, *CEP290*-related IRDs were predominantly associated with severe, early-onset, and progressive retinal degeneration, typically manifesting in childhood [7,8,9]. However, recent studies have revealed a broader phenotypic spectrum of *CEP290* variants than was previously understood [11,12]. While many cases follow severe disease trajectories, some individuals have exhibited milder, more slowly progressing forms of retinal dystrophy [13,14]. These findings highlight the complexity of genotype–phenotype correlations and stress the need for individualized approaches to clinical care. Although *CEP290*-related CRD has been previously documented [13], the variability in its clinical presentation within families remains underreported.

In this study, we describe two siblings with biallelic *CEP290* variants who exhibit markedly different forms of non-syndromic CRD, ranging from severe to notably milder, slowly progressive presentations. By characterizing this variability, we aim to expand the understanding of the phenotypic spectrum of *CEP290*-related IRDs and to emphasize the importance of tailored clinical care strategies.

## 2. Materials and Methods

### 2.1. Case Presentation

The two siblings described in this study are part of an internal cohort of 1167 individuals, primarily of Spanish descent, diagnosed with IRDs (internal data). This cohort includes both pediatric and adult cases gathered through the Hereditary Retinal Dystrophies Unit—a collaborative unit integrating the Ophthalmology departments at SJD Barcelona Children’s Hospital and Bellvitge University Hospital. Of these individuals, 964 underwent whole-exome sequencing (WES) as part of the genetic screening program. The cohort represents a wide range of IRD phenotypes, including both syndromic and non-syndromic forms, such as RP, LCA, CRD, SLSN, and JBTS.

### 2.2. Ophtalmologic Evaluation

Both probands underwent comprehensive ophthalmological assessments. Best-corrected visual acuity (BCVA) was measured using the ETDRS chart, and biomicroscopy was performed using slit-lamp examination. Tropicamide was administered to dilate the pupils, enabling detailed fundus examination. Retinal imaging, including fundus autofluorescence (FAF) and retinography, was obtained using Optos ultra-widefield (UWF™) retinal imaging (Optos California, Optos plc). To assess the retinal layers and detect structural abnormalities, swept-source optical coherence tomography (OCT) (Topcon Triton, Topcon Corporation) was performed. Retinal function was evaluated through central visual field testing (CV 10.2) and electrophysiology (Roland Consult), including full-field electroretinography (ffERG) and multifocal electroretinography (mfERG), following the current standards of the International Society for Clinical and Electrophysiology of Vision (ISCEV).

### 2.3. Whole-Exome Sequencing and Data Analysis

WES was conducted using the SureSelect XT HS Low Input Human All Exon v8 kit (Agilent Technologies, Inc., Santa Clara, CA, USA). Paired-end sequencing (2 × 150 bp) was performed on the NovaSeq X Plus System (Illumina, San Diego, CA, USA). Bioinformatic analysis was performed using the Data Genomics Exome pipeline (v22.4.0) developed by Health in Code (Valencia, Spain). Copy number variation (CNV) analysis was carried out using VarSeq (Golden Helix, Inc., Bozeman, MT, USA). Sequence alignment and variant calling were conducted against the hg19 human genome reference.

A virtual panel including 489 genes associated with IRDs was applied to filter variants (Supplementary Data). A minimum read depth of 20 and an allele frequency threshold of 20% were applied. Variants with a minor allele frequency higher than 1 in 500, according to gnomAD v2.1.1 [15], were excluded. Missense variants were analyzed using REVEL [16], while splice variants were assessed with SpliceAI [17]. Protein–protein interactions were analyzed with the STRING database [18]. Variants were classified following ACMG-AMP guidelines [19] and SVI-WG recommendations [20]. Parental DNA was analyzed using Sanger sequencing to confirm inheritance in *trans*.

Ethical approval for this study was obtained (reference number PR235/24) from the Research Ethics Committee of Bellvitge University Hospital. Informed consent was secured for genetic testing and the publication of clinical data.

### 2.4. mRNA Analysis of CEP290 Variants

To investigate the potential impact of the variants on mRNA, multiple experiments were conducted. mRNA was extracted from fresh whole blood samples using the Maxwell^®^ RSC simplyRNA Blood Kit (Promega, San Diego, CA, USA) following the manufacturer’s protocol. Complementary DNA (cDNA) synthesis was performed using the PrimeScript^TM^ RT Reagent Kit (Takara Bio Inc., Kusatsu, Japan).

### 2.4.1. Real-Time Quantitative PCR (RT-qPCR) Analysis

Gene expression analysis was performed in triplicate using TaqMan^®^ probes Hs01551628_m1 (targeting exons 5–6 of the *CEP290* gene) and Hs99999905_m1 (targeting exon 3 of *GAPDH*) (Applied Biosystems, Foster City, CA, USA). The relative fold change in gene expression was calculated using the 2^–∆∆Ct^ method, as described by Livak et al. (2001) [21].

### 2.4.2. Exon Skipping Analysis

To evaluate the impact of the identified variants on mRNA splicing, primers were designed to amplify regions including exons 10–14 and exons 40–44 of the *CEP290* gene (NM_025114.3) (Appendix A). These regions were amplified under standard PCR conditions, and the resulting amplicons were purified using the ExoSAP-IT^TM^ (Applied Biosystems, Foster City, CA, USA) according to the manufacturer’s instructions. Purified PCR products were sequenced using the BigDye^TM^ Terminator v3.1 Cycle Sequencing Kit (Applied Biosystems). The resulting electropherograms were analyzed with Mutation Surveyor v5.1.2 software (SoftGenetics, LLC, State College, PA, USA).

## 3. Results

### 3.1. Clinical Findings

Proband 1 (Figure 1) is a 51-year-old male diagnosed with RP at age 3, initially presenting with photophobia. His height was measured at 174 cm (50th percentile), weight was 81.5 kg, and head circumference was 56 cm (50th percentile), with no dysmorphic features or non-ophthalmological health concerns. His symptoms included worsening night blindness and difficulty adjusting to light variations. At age 50, his ophthalmic examination revealed severe central visual impairment (20/500 OD, 20/640 OS) and peripheral visual field defects. Anterior segment examination showed bilateral central subcapsular cataract with intraocular pressure (IOP) of 20 mmHg. Fundus examination showed optic disc pallor, extensive pigmentary changes with bone spicules, macular atrophy, and thinned retinal vessels. FAF demonstrated hypoautofluorescence in the peripheral retina, extending into the superior and inferior nasal quadrants, and a hyperautofluorescent foveal ring (Figure 2A). OCT showed outer retinal layer loss and retinal pigment epithelium (RPE) thinning (Figure 2B). These findings led to a clinical diagnosis of early onset non-syndromic CRD.

Proband 2 (Figure 1), the 55-year-old sister of proband 1, was referred at age 51 due to eye fatigue. Her height was measured at 152 cm (10–25th percentile), weight was 56.5 kg, and head circumference was 55 cm (50th percentile), with no dysmorphic features or additional health concerns. She exhibited milder ophthalmological symptoms, including mild photophobia and peripheral visual field loss, especially in the lower and right lateral areas, but denied experiencing nyctalopia. Upon inquiry, childhood difficulties with object tracking were retrospectively reported. During the initial evaluation, her ophthalmic examination showed moderate visual impairment (20/100 OD, 20/50 OS) and bilateral posterior subcapsular cataracts. Fundus examination revealed less pronounced bilateral pigmentary changes and bone spicules in the inferior and temporal peripheral retina compared to proband 1, with normal optic disc and retinal vasculature. FAF revealed localized bilateral hypoautofluorescent patches in the inferior and temporal peripheral retina, sparing the posterior pole with a hyperautofluorescent demarcation line (Figure 2C). OCT showed minimal structural abnormalities with slight outer retinal layer disruption and preserved inner retina and foveal contour (Figure 2D). ffERG showed delayed and reduced photopic responses, preserved scotopic responses, and abolished flicker responses. MfERG revealed reduced central amplitudes (Appendix A). These findings indicated a milder form of non-syndromic CRD.

### 3.2. Identification of Biallelic Variants in CEP290

WES analysis identified two heterozygous variants of *CEP290* in proband 1: NM_025114.3:c.955del, p.(Ser319LeufsTer16) and NM_025114.3:c.5777G>C, p.(Arg1926Pro). Due to the mild clinical presentation observed in proband 2, an initial diagnosis of *CEP290*-related IRD was considered unlikely. Consequently, WES was performed on proband 2, rather than targeted Sanger sequencing of the identified variants, to explore other potential molecular causes. Unexpectedly, the same two *CEP290* variants identified in proband 1 were also present in proband 2, suggesting a shared genetic basis for their IRD diagnosis, despite the observed clinical differences. No additional pathogenic/likely pathogenic variants or CNVs were detected in either sibling.

The variant NM_025114.3:c.955del, located in exon 12/54, resulted in a nucleotide deletion predicted to cause a frameshift, leading to a premature stop codon, p.(Ser319LeufsTer16). This variant was predicted to result in a truncated protein, disrupting the coiled-coil domain (amino acids 59–565), which is critical for maintaining the proper conformation of the CEP290 protein. More likely, it would cause a loss of function through mRNA degradation via nonsense-mediated decay (NMD), a mechanism strongly associated with pathogenicity in this gene (PVS1-very strong). Its low frequency in the gnomAD v2.1.1 population database (0.0006149%) (PM2-supporting) aligned with the recessive inheritance model for *CEP290*-related disorders. This variant was classified as pathogenic in ClinVar (ID:1073010) and HGMD (CD1212112). It has been associated with MKS in a homozygous patient [22] and with LCA in a compound heterozygous patient carrying another pathogenic variant (PM3-moderate) [23]. Additional truncating variants in *CEP290* have been linked to various ciliopathies, including JBTS and Bardet–Biedl syndrome [8,9,11]. Based on current evidence and ACMG classification guidelines, this variant has been classified as pathogenic.

The variant NM_025114.3:c.5777G>C, located in exon 42/54, resulted in an amino acid substitution, p.(Arg1926Pro), changing a positively charged arginine to a neutral proline. This protein codon resided within the RepA/Rep protein KID domain, a region essential for protein kinase interaction. It has a low frequency in the gnomAD v2.1.1 population database (0.0004187%) (PM2-supporting), consistent with its association with *CEP290*-related disorders in a recessive inheritance model. In silico predictions for this variant were uncertain (REVEL score 0.35), and functional studies have not been conducted. It has been classified as pathogenic in various databases, including HGMD (CM111853), ClinVar (ID: 659046), and LOVD (CEP290_000296). This variant has been reported in individuals with LCA in compound heterozygosity with other pathogenic/likely pathogenic variants (PM3-strong) [4,8,10,24], and it has been shown to segregate in a family with two affected members (PP1-supporting) [10]. Based on current evidence and following ACMG classification guidelines, this variant has been classified as likely pathogenic.

Parental segregation analysis confirmed that these variants were inherited in *trans* configuration.

### 3.3. Characterization of Identified CEP290 Variants and Their Impact on Splicing

RT-qPCR results showed a significant reduction in *CEP290* mRNA levels in both proband 1 (*p* = 0.0004) and proband 2 (*p* = 0.004) compared to the control sample (Appendix A).

These findings suggest that the frameshift variant c.955del partially induced NMD, as the expression level was not reduced by 50%, indicating incomplete NMD activation. Although mRNA expression levels varied slightly between the two probands, the differences are not statistically significant.

To evaluate whether the c.955del variant caused exon skipping, a mechanism previously associated with milder forms of the disease [25], Sanger sequencing of exons 10–14 was performed (Appendix A). The results confirmed that the c.955del variant did not trigger exon 12 skipping, consistent with in silico predictions (SpliceAI score = 0.04). Nonetheless, allele ratio analysis revealed a reduced presence of the mutant allele (45%) compared to the normal allele (55%). These findings support the conclusion of incomplete NMD activation, consistent with the observed *CEP290* mRNA expression results.

Sanger sequencing of exons 40–44 confirmed the presence of the missense variant c.5777G>C, with no evidence of exon 42 skipping, aligning with in silico predictions (SpliceAI score = 0.03). Interestingly, the results revealed exon 41 skipping in both the probands and the control samples. Notably, the agarose gel analysis (Appendix A) showed more intense bands corresponding to exon 41 skipping in the probands’ samples compared to the control. Additionally, a third band, approximately 350 bp in size, was detected. Future studies should aim to sequence this band to evaluate its potential impact on *CEP290* function.

## 4. Discussion

This study highlights the significant intra-familial variability in *CEP290*-related IRDs, demonstrated by the contrasting phenotypes observed in two siblings carrying identical biallelic *CEP290* variants. Proband 1 exhibited a severe, early-onset CRD with rapid disease progression, while proband 2 presented a milder, late-onset form with slower progression. Comprehensive evaluations by nephrology and neurology affirmed the non-syndromic nature of their conditions.

Establishing clear genotype–phenotype correlations in *CEP290*-related conditions remains challenging. Different *CEP290* variants associated with CRD have been documented, yet most cases typically describe severe, early-onset presentations [6,13,26]. This study contributes novel insights by documenting the coexistence of a mild, late-onset CRD phenotype alongside a severe phenotype within the same family, emphasizing the complexity of the genetic and non-genetic factors influencing disease expression [26].

The c.955del frameshift variant has been associated with severe phenotypes, such as MKS and LCA, when combined with other loss-of-function alleles [4,6]. Meanwhile, the c.5777G>C missense variant has been reported in both homozygosity and compound heterozygosity, with null variants in severe cases [8,10,24]. Notably, to our knowledge, this study is the first to document the coexistence of c.955del with a missense variant, potentially explaining the milder phenotype observed in proband 2. These findings emphasize the critical role that specific variant combinations play in shaping genotype–phenotype correlations. For example, the c.2991 + 1655A>G deep intronic variant, first described by den Hollander et al. (2006) [27], has been shown to preserve correct splicing in most tissues but is insufficient in photoreceptors, leading to retinal-specific manifestations [11,28].

Residual CEP290 protein activity, alternative splicing, genetic modifiers, and environmental factors likely influence the phenotypic differences between the siblings [29,30,31].

The observed reduction in CEP290 expression is consistent with the activation of NMD. However, further experiments using NMD inhibitors would be necessary in order to confirm this hypothesis. Additionally, mechanisms like nonsense-associated altered splicing (NAS) or basal exon skipping (BES) could generate functional transcripts that bypass premature termination codons [11,25], potentially resulting in residual protein expression and a milder phenotype [32]. Exon 41 skipping was observed in both siblings and in the control sample, suggesting that it may be associated with the variant identified in exon 42 (c.5777G>C), given that changes in exon donor or acceptor sites can impact the splicing of neighboring exons, as shown in prior studies [11,25]. However, this exon skipping was also observed in all samples and controls in the study conducted by Drivas et al. (2015) [33].

Genetic modifiers such as *AHI1*, *RPGRIP1L*, and *CC2D2A*, which interact with *CEP290* at the protein level, may also contribute to phenotypic variability (Appendix A) [34,35]. Proband 1 carried more variants in these genes than proband 2 (Appendix A), including a *CC2D2A* variant predicted to have deleterious effects on protein level (Revel score: 0.683), which could partially explain the more severe phenotype.

Environmental and epigenetic factors, such as mechanisms involving endoplasmic reticulum-associated protein degradation (ERAD), may further influence disease severity [36,37]. These non-genetic influences, combined with the unique structural and functional demands of retinal cilia [3], highlight the complexity of *CEP290*-related retinal diseases.

Within our internal cohort of 964 IRD cases, nine additional individuals with biallelic *CEP290* pathogenic variants were identified (Appendix A), most of whom exhibited severe, early-onset retinal degeneration consistent with previously documented phenotypes [12]. Only one individual carried a missense variant in combination with a loss-of-function variant. While this patient displayed a non-syndromic phenotype, their disease course was far more aggressive than that of proband 2, further emphasizing the complexity of *CEP290* genotype–phenotype relationships.

Emerging therapeutic approaches, such as gene editing and antisense oligonucleotide (ASO) therapies, hold promise for managing *CEP290*-related IRDs [38,39]. However, the significant phenotypic variability associated with *CEP290* highlights the need for personalized strategies that account for specific genotypes, splicing profiles, and modifier influences to maximize treatment efficacy.

In conclusion, this study broadens the understanding of *CEP290*-related IRDs by documenting the coexistence of both severe and mild phenotypes within the same family. It highlights the complex interplay of the genetic, molecular, and environmental factors that contribute to the variability in disease expression. Although preliminary findings on splicing alterations were explored, they serve as a foundation for future, more comprehensive investigations into *CEP290* expression and splicing mechanisms.

## Figures and Tables

**Figure 1 genes-15-01584-f001:**
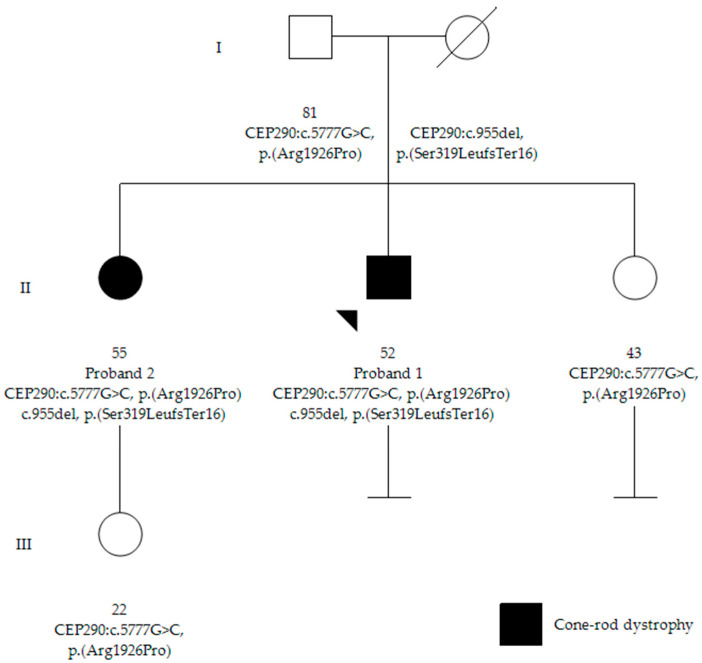
Family pedigree. Proband 1 is indicated by an arrow, and individuals with a clinical diagnosis of cone-rod dystrophy are shaded in black. Proband 2 has a 22-year-old asymptomatic daughter, and the affected siblings share a third sister who underwent a normal ophthalmological examination and remained unaffected. The family is of Spanish origin, with no other reported cases of ophthalmological diseases and no known consanguinity among relatives.

**Figure 2 genes-15-01584-f002:**
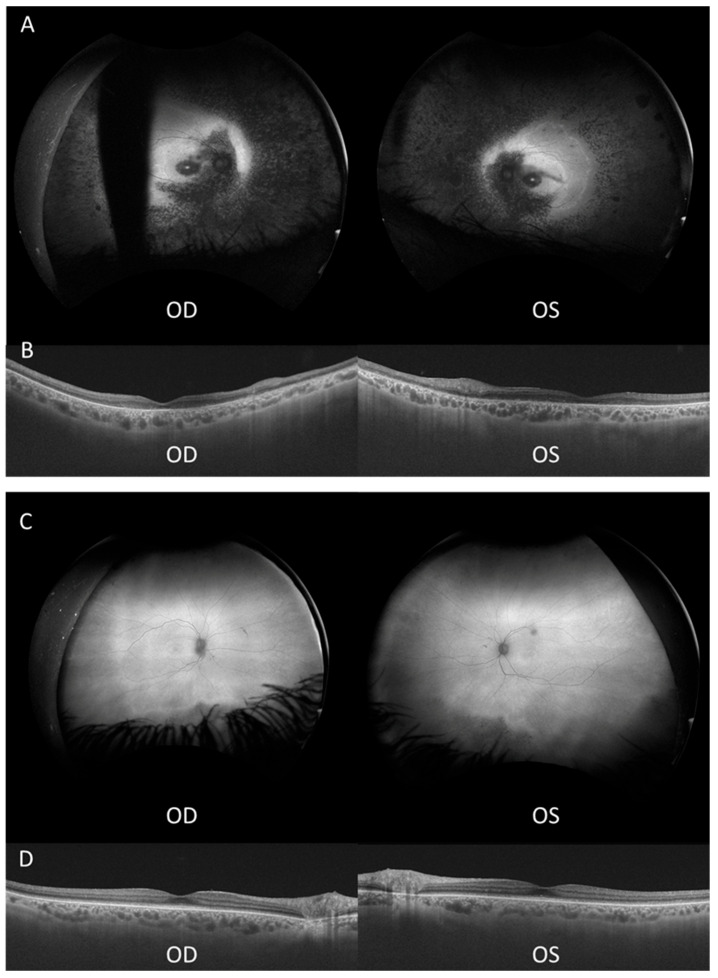
Ultra-widefield autofluorescence (FAF) and optical coherence tomography (OCT) of both eyes in each sibling. (**A**) FAF image of proband 1, with blinking artifact in OD. (**B**) OCT of proband 1. (**C**) FAF image of proband 2. (**D**) OCT of proband 2 showing discontinuities in external retinal layers. OD, right eye; OS, left eye.

## Data Availability

The original contributions presented in the study are included in the article/Appendix A, further inquiries can be directed to the corresponding authors.

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
