# Peer review of "Expanding the Clinical Spectrum of CEP290 Variants: A Case Report on Non-Syndromic Retinal Dystrophy with a Mild Phenotype"

_genes, 2024, doi:10.3390/genes15121584_

Round 1

Reviewer 1 Report

Comments and Suggestions for Authors

The authors reported two cases from the same family of CRD associated with the CEP290 gene, which is more frequently associated with LCA or severe syndromic retinal dystrophies than with late and moderate forms.

The manuscript is clear and well written.

I have some remarks:

In the material and methods, it would be necessary to specify the number of cases analyzed with this method (WES). It is said in the discussion "950 patients", but it would be necessary to indicate it further upstream. Similarly, do all the patients in this initial cohort have LCA, if not, how much?

-Were CNVs searched by WES?

-This is not the first description of patients with CEP290-associated CRD; there are 5 other variants reported in HGMD. This study is therefore not novel.

-To confirm the involvement of these variants in the pathology, additional clinical examinations would be required : MRI to look for the molar tooth sign, renal exploration... because for now, to my knowledge, there is only one variant involved in non-syndromic forms: c.2991+1655A>G

Reviewer 2 Report

Comments and Suggestions for Authors

Esteve Garcia et al. report on a case of non-syndromic retinal dystrophy with mild phenotype.

Major changes requested: in clinical reporting, anthropometric data (height, weight and head circumference, and centiles ) should be added to stress the "non-syndromic"status of the patients;

similarly some sentences should be added to stress that no dysmprphisms or any other relevant non ocular disease is present.

3.2 : lines 136 and 143: it should be said more clearly that no other pathogenic variant was found in the panel analyzed for both sibs 

the description of the variants found  is OK.

lines w00-205: if I understand well, you suggest that levels of residual activity may explain the clinical differences; if so you should modifie the sentence saying something like: "unexplained differences of residual activity in the two sibs  may explain the clinical differences" 

line 206: the observation about possible effect of other genes is a good one, but........you have all WES data, why not to check for other pathogenic variants outside the panel analyzed? or why not to check, at least,  for variants in genes interacting with your gene ? (see the STRING database) 

the best would be to do this further analysis, or at least discuss why it was not done,

in general a short comment on the relevance of the exon carrying the mutation on protein function and structurecould also be added

Minor comment: pedigree drawing in Fig. 1 should be improved: you can cancel non necessary subjects and use a specific program to obtain a better image 

line 230 you could state  which type of variant was found in these patients (LOF, missense ...)

line 225 you report data about your very large cohort of patients with IRD; I guess they are (all?) Spanish.  a comment about ethnic origin of the family could be added, as well as , if available, data about regional incidence of the two found variants in Spain and relation to the city of origin of the parents. 

Round 2

Reviewer 1 Report

Comments and Suggestions for Authors

The authors improved their manuscript and have taken into account reviewer's comments
